# Post-Harvest Processing Methods Have Critical Roles in the Contents of Active Ingredients of *Scutellaria baicalensis* Georgi

**DOI:** 10.3390/molecules27238302

**Published:** 2022-11-28

**Authors:** Liuwei Zhang, Xuemei Zhang, Zongsuo Liang

**Affiliations:** 1College of Chemistry and Pharmacy, Northwest A&F University, Xianyang 712100, China; 2Shandong (Linyi) Institute of Modern Agriculture, Zhejiang University, Linyi 276000, China; 3College of Life Science and Medicine, Zhejiang Sci-Tech University, Hangzhou 310018, China; 4College of Life Science, Northwest A&F University, Xianyang 712100, China

**Keywords:** *Scutellaria baicalensis* Georgi, fresh processing, drying method, active components

## Abstract

To find the best post-harvest processing method for Scutellaria baicalensis Georgi, we explored the effects of fresh and traditional processing on the active ingredients in *S. baicalensis* and evaluated three drying techniques to determine the optimal post-harvest processing technique. We quantified four active ingredients (baicalin, baicalein, wogonoside, and wogonin) in 16 different processed *S. baicalensis* samples that were harvested from Tongchuan, Shaanxi province, by HPLC (high-performance liquid chromatography). In addition, we performed a similarity analysis (SA), a hierarchical cluster analysis (HCA), and a principal component analysis (PCA) on the common peaks in *S. baicalensis* that were identified by the HPLC fingerprints. Compared to the traditional processing method, the fresh processing method could better preserve the four active ingredients in *S. baicalensis*, meanwhile, the similarity analysis (0.997–1.000) showed that the fresh processing was more similar to the traditional processing, and it did not change the type of 18 active ingredients in *S. baicalensis*. The cluster analysis results showed that the shade drying and sun drying methods results were more similar to each other, while the oven drying (60 °C) method results were clustered into one category. According to the results of the principal component analysis, S9, S7, and S8 had higher scores, and they were relatively well processed under these processing settings. Fresh processing could be an alternative to traditional processing; the moisture content was reduced to 24.38% under the sun drying condition, and it was the optimal post-harvest processing solution for *S. baicalensis*.

## 1. Introduction

Medicinal plants are one of the components of global medicine. Huang Qin is the dried roots of *Scutellaria baicalensis* Georgi, a well-known botanical in the family Labiatae (Figure 1). It has the qualities of clearing heat and drying moisture, halting bleeding, and relaxing the fetus [1]. Several traditional Chinese medicine prescriptions contain *S. baicalensis*, such as the Huangqin Qingre Chubi Capsule, the San Wu Huangqin Decoction, the Sanhuang Tablet, and the oral dose of Shuanghuanglian [2,3,4,5]. In recent medical research, *S. baicalensis* exhibits anti-inflammatory, anti-tumor, anti-bacterial, hypotensive, and anti-depressant properties [6,7]. Especially during the fight against COVID-19, *S. baicalensis* played an anti-viral role [8]. The active ingredients in the roots of *S. baicalensis* are mainly flavonoids, alkaloids, polysaccharides, triterpenoids, diterpenes, iridoid glycosides, essential oils, and other compounds. In particular, flavonoids are one of the most important groups of active ingredients, including baicalin, baicalein, wogonoside, and wogonin [9,10,11,12]. These flavonoids are a series of secondary metabolites that are synthesized in plants via the shikimate pathway (Figure 2). The biosynthesis process is regulated by various enzymes, and it is susceptible to external factors such as the environment [13,14]. Modern pharmacological experiments have demonstrated that baicalein, wogonin, and pyroxylin A showed potent anti-H1N1, cytotoxic, and Nrf2 activation activities [15]. Baicalin has anti-Parkinson’s disease activity [16], anti-inflammatory and anti-oxidant [17], neuroprotection properties [18,19]. Baicalin is the main active component of *S. baicalensis*, accounting for around 0.1 *g*/*g* (ratio is in raw material) [20,21]. In some pieces of literature, baicalin is considered to be one of the main indicators of the quality of *S. baicalensis* [22]. However, baicalein has a wider absorption window than baicalin does, for example, the degree of absorption of baicalein in all of the parts of the stomach and intestine is significantly higher than that of baicalin. Baicalin is only moderately absorbed in the stomach, and it is largely unabsorbed in the small intestine and colon; baicalein is well absorbed in the stomach and small intestine, and to some extent, in the colon, and it can exert better pharmacological effects [23]. It should not be overlooked. Therefore, for the quality assessment of *S. baicalensis*, a comprehensive evaluation of Baicalin, Wogonoside, Baicalein, and Wogonin should be carried out.

Since the 21st Century, the use of *S. baicalensis* has gradually increased, leading to a shortage of wild *S. baicalensis* resources, and the artificial cultivation of *S. baicalensis* has begun to replace wild *S. baicalensis* as the main source of *S. baicalensis* [24]. The cultivation and post-harvest processing of *S. baicalensis* are also becoming hot research topics. Because of the high moisture content after harvesting, it will easily become moldy and it will decay if it is not processed in time, which can seriously affect the quality of the herbs, its clinical efficacy, and its economic efficiency. The Traditional Chinese medicine (TCM) post-harvest processing includes cleaning, drying, and cutting [25]. Many factors control the post-harvest quality. For instance, the microbial activity, moisture content, room temperature, harvest time, and light affect the quality and stability of *S. baicalensis* Georgi [26,27,28,29]. One of the most influential factors is the drying method. The benefits from drying include controlling the microbial activity and enabling its long-term storage while maintaining its potency, taste, and medicinal properties [30,31]. In recent years, with the development of technology, the drying methods have become more and more diverse, while the existing Chinese Pharmacopoeia has no uniform standards for the processing methods of the TCMs. Some enterprises rely heavily on experience to process TCMs, causing the quality of the TCMs to fluctuate. In addition, due to the lack of sophisticated drying equipment, it is impossible to meet the TCMs’ mass manufacturing demand. Consequently, searching for low-cost, simple, and convenient drying methods with quality assurance is a challenge that is faced by the enterprises at present, as well as it being a research hotspot that many research institutions are worried about.

In the traditional post-harvest processing of *S. baicalensis*, the roots need to be dried twice (a first drying step before the steam softening, and a second one after the slicing), which is both energy and time-consuming. Therefore, it is necessary to optimize the traditional processing of *S. baicalensis* to minimize the energy consumption during the drying process and minimize the steam softening process. One prospective strategy suggested that the fresh samples were cut into slices and processed after their harvest in the original cultivated area, thus saving time and effort, and maintaining the quality of the herbs while processing [32]. At the same time, the Protection and Development Plan of Chinese Medicinal Materials (2015–2020) also clearly states that further research is to be focused on deep processing technologies for the fresh cutting of TCMs. At present, several herbs have been fresh processed in the original cultivated area, such as *Salvia miltiorrhiza* Bunge and *Angelica sinensis* Radix [33,34]. However, some studies on the post-harvest processing of *S. baicalensis* have mostly focused on the effects of different drying methods, such as shade drying and sun drying, on active ingredients [35].

Our study aimed to explore the feasibility of replacing the traditional processing methods for the fresh processing of *S. baicalensis* and to also find the best drying method for *S. baicalensis* by using the high performance liquid chromatography (HPLC) method to determine the contents of the active ingredients and evaluate the overall quality of differently processed *S. baicalensis* using combining fingerprinting, a similarity analysis (SA), a hierarchical cluster analysis (HCA), and a principal component analysis (PCA). Collectively, the above results will provide a theoretical basis for the scientific post-harvest processing in the producing area of *S. baicalensis*.

## 2. Results

### 2.1. The Relationship between Different Drying Methods and Moisture Content

As observed in Figure 3, the moisture content of the samples treated using the oven drying method (60 °C) reached below 20% within one day. However, the moisture content decreased very slowly for the sun drying and shade drying methods. It took 6 and 10 days, respectively, for the moisture content to reach approximately 30%.

The oven drying (60 °C) method has a relatively high temperature compared to the shade drying and sun drying methods. In addition, the oven drying (60 °C) method has a relatively stable heating environment, in relation to its temperature, humidity, and light, so the moisture content of the sample dried by the shade drying and sun drying methods decreased more slowly.

### 2.2. Effects of Fresh and Traditional Processing Methods on the Contents of the Main Active Ingredients

In combination with Figure 4A and Table 1, for the samples (S2, S3, S4, and S5) that were processed by fresh processing, the content of baicalin is the highest in S4. However, its content was still reduced by 6.30% compared to the content of S14 that was processed by traditional processing. The content of Wogonoside in the samples S3, S4, S5 was not significantly different from that in S14. The contents of baicalein and wogonin in the S1 were 6.66 and 4.48 times higher than they were in S14, respectively. It can be seen in Figure 4B and Table 1 that the content of baicalin was the highest in S9, which was not significantly different from S15; however, the content of baicalin tended to increase as the moisture content gradually decreased. The content of wogonoside was the highest in S7, where it was 6.90% higher than it was in S15. There was no significant difference between S9 and S15. The contents of baicalein and wogonin were significantly higher than those of the traditional process. The highest contents in the S1 were 3.96 and 1.37 times higher than they were in S15. It can be seen in Figure 4C and Table 1 that the content of baicalin in the sample processed by fresh processing was significantly higher than it was in S16. The content values of baicalin, wogonoside, and baicalein in S1 were 0.08, 0.14, and 0.26 times higher than those in S16, respectively. The content of wogonin in the samples (S10, S11, S12, S13) that were processed by oven drying (60 °C) was lower than it was in S16.

Compared to the traditional processing method, there was no difference in the content of baicalin and wogonoside in the samples dried by sun drying when the moisture content reached 24.38% during the fresh processing; however, for the content of baicalein and wogonin, the fresh processing was superior to the traditional method, and S1 (no drying) can be considered to be the most advantageous method for baicalein. In the samples processed by oven drying (60 °C), fresh processing was more favorable for preserving baicalin, wogonoside, baicalein, but it was less favorable for preserving wogonin. In the samples processed by the shade drying method, fresh processing was not favorable for preserving baicalin.

### 2.3. Differences in the Contents of the Four Active Ingredients among Different Drying Conditions in Fresh Processing Method

According to the trend in Figure 5A, for the content of baicalin, shade drying (S6–S7–S8–S9) > sun drying (S2–S3–S4–S5) > oven drying (60 °C) (S10–S11–S12–S13). For the oven drying (60 °C) method, S1–S10–S11–S12–S13 showed a fluctuating decrease, and the content of baicalin in S1 was greater than that it was in S10, S11, S12, and S13.

For the content of wogonoside in the sample (Figure 5B), the sun drying method (S6–S7–S8–S9) and the shade drying (S2–S3–S4–S5) method were better than the oven drying method was (60 °C) (S10–S11–S12–S13). As the moisture content decreased, the content of wogonoside increased from S1–S6–S7–S8–S9 for the sun drying method, and it remained stable from S1–S2–S3–S4–S5 for the shade drying method. However, the content of wogonoside firstly decreased, then, it increased from S1–S11–S12–S13–S14, but the content values of wogonoside in S10, S11, S12, and S13 were still smaller than it was for S1. For the content of baicalein and wogonin in the samples (Figure 5C,D), the oven drying (60 °C) method (S10–S11–S12–S13) is better than the shade drying method (S2–S3–S4–S5) and the sun drying method are (S6–S7–S8–S9). Additionally, except for the content of wogonin in S10 which was greater than it was in S1, the content of baicalein and wogonin in the samples conducted by the sun drying method (S6–S7–S8–S9), the shade drying method (S2–S3–S4–S5), and the oven drying method (60 °C) (S10–S11–S12–S13) all were lower.

The three drying methods have their advantages and disadvantages. In summary, the sun drying and shade drying methods resulted in a decrease in the contents of baicalein and wogonin and an increase in the contents of baicalin and wogonoside. However, the sun drying method showed a more significant increase in the content of baicalin and wogonoside compared to the shade drying method. The oven drying (60 °C) method resulted in a decrease in the contents of baicalin, wogonoside, baicalein, and wogonin, but the oven drying (60 °C) method results in less loss in the content baicalein and wogonin compared to the other two drying methods.

### 2.4. HPLC Fingerprinting and SA of S. baicalensis Treated by Different Processing Methods

There were 18 peaks marked more than 90% of the total peak area (Appendix A). After adding the standards and comparing the retention times, four peaks were identified (Figure 6). They correspond to: peak 7—baicalin; peak 13—wogonoside; peak 16—baicalein; peak 18—wogonin.

The similarity values of the 16 batches of samples with the reference fingerprint ranged from 0.997 to 1.000 (Appendix A). Moreover, the RSD values of the relative peak area of the common peaks were all greater than 5% (Appendix A). The RSD values of the relative retention time of each common peak ranged from 0.526% to 1.352% (Appendix A) when we were using common peak 13 as the reference peak. It indicates that the category of the compositions was the same in the roots of *S. baicalensis* processed by different treatments, but the content varies greatly.

### 2.5. HCA of S. baicalensis Treated by Different Processing Methods

As shown in Figure 7, the S7, S8, S9, S15 and S2, S5, and S14 samples were assembled into the first group sample; the S10, S11, S12, S13, and S16, and S1 ones were assembled into the second group, while the S3 and S4 ones were assembled into the third group, and S6 was assembled into the fourth group. The drying temperature was higher in treatment group 2 than it was in groups 1, 3, and 4. It is concluded that temperature was considered to be one of the main factors influencing the post-processing of *S. baicalensis*.

### 2.6. PCA of S. baicalensis Treated by Different Processing Methods

The ranking is shown in Table 2, and it can be obtained that S9 that was dried by the sun until the moisture content reached 24.38% was the best, which was followed by S7 (43.22%) and S8 (31.61%). As shown in the score plot of the PCA results (Figure 8), S9 was assembled into the first group; S1, S3, S4, S5, S7, S8, S10, S13, and S15 were assembled into the second group; S2, S6, S14, and S16 were assembled into the third group samples; S11 and S12 were assembled into the fourth group samples.

## 3. Discussion

### 3.1. For Post-Harvest Processing of S. baicalensis, Fresh Processing Can Replace Traditional Processing

In this study, it was found that under certain drying conditions and moisture contents, the content of the overall composition in the roots of *S. baicalensis* conducted by fresh processing in the original cultivated area was better than or equal to that of S. baicalensis that was processed under traditional processing conditions, such as the sample dried by sun drying method to a moisture content of 24.38%, 31.61%, and 43.88%, etc. Therefore, it has been demonstrated that fresh processing in the original cultivated area can replace the traditional processing method. Other studies also obtained similar experimental results [36,37]. The content of active ingredients in *S. baicalensis* that was processed by fresh processing was more than 50% higher than that of *S. baicalensis* that was processed by traditional methods. In addition, the SA of the HPLC fingerprints of the 18 components between the traditional and fresh processing indicated that the fresh processing did not change the type of active ingredients compared to the traditional processing. In addition, fresh processing in the original cultivated area saves processing costs, time, and energy, while maintaining the quality of the TCMs [38,39]. Especially for the traditional processing of *S. baicalensis*, the softening process is added to make the dried roots of *S. baicalensis* more easy to cut, and the softening methods include water steam softening and boiling them in water. Similar experimental results have proven that the softening process leads to the loss of some of the glycosides. For example, during the traditional processing of *Angelica sinensis*, the softening process causes the degradation of the glycosides [40].

Meanwhile, the enzymes take longer to function due to the relatively long duration of the traditional processing. However, this problem was effectively solved by the fresh processing method. After *S. baicalensis* underwent the fresh processing in the original cultivated area, one of the necessary conditions for enzymatic activity was destroyed due to the quick reduction in the moisture content, and baicalin and wogonoside could be relatively well preserved. In the future, fresh processing in the original cultivated area will become a new energy-saving method instead of traditional processing.

### 3.2. Effect of Different Drying Methods on the Active Ingredients of S. baicalensis

The temperature and time are two main factors affecting the drying method. Of the three drying methods, oven drying (60 °C) is accompanied by a relatively high temperature, while the shade drying method requires a relatively longer drying time. This was confirmed in the present experiment. It takes 13 days to reduce the moisture content of the samples that were processed by the shade drying method to less than 30%, which takes much longer than the oven drying (60 °C) and sun drying methods do. The phenomenon resulted in a lower baicalin content in the samples that were dried in the shade than for those that were dried in the sun. The possible explanations for this phenomenon are that baicalin is immediately hydrolyzed to baicalein by the action of the endogenous enzyme glucuronidase (GUS), which is active for a more extended period under shade drying conditions [41,42,43,44].

The roots of *S. baicalensis* are physiologically active for some time after the harvesting, and prolonged high-temperature stress (40 °C) inhibits the activity of phenylalanine ammonia-lyase (PAL), and consequently, this affects the accumulation of baicalin [45]. The content of baicalin and wogonoside is significantly reduced in the samples that were dried in the oven (60 °C). During the oven drying (60 °C) process, the internal heat could not be dissipated over time with the extension of the drying time, resulting in continuous heating inside the roots. That creates robust conditions for a series of complex enzymatic oxidation, polymerization, and other biochemical reactions. Some studies have shown that the enzymatic (GUS) reaction of baicalin and other glycosides is accelerated in a hot and humid environment, resulting in the enzymatic degradation of baicalin and wogonoside into baicalein and wogonin, which contain phenolic hydroxyl groups in adjacent positions and are easily oxidized into quinone derivatives, and they appear green [46,47]. In addition, many phenolic and alcoholic hydroxyl groups in the flavonoid structure are unstable, and they are easily destroyed at high temperatures. This study has come to the same conclusion [48].

In the study, we found that the oven (60 °C) drying method is more favorable for preserving baicalein and wogonin than the sun-drying and shade drying methods are. It has been noted that baicalin and wogonoside are primarily found in the cortical layer; however, baicalein and wogonin are mainly distributed in the cortical layer and xylem [49]. Compared to the cortical layer, the xylem part is more susceptible to external environmental influences, including temperature, humidity, and light, so baicalein and wogonin may be damaged. It has also been documented in the other study that baicalein is more affected by light, but it is less affected by temperature [50]. Compared with the drying method, the sun drying and shade drying methods are more susceptible to the influence of light, temperature, humidity, and other external environments, so the shade drying and sun drying methods are relatively more unfavorable to the preservation of baicalein and wogonin.

### 3.3. Different Quality Controls for Components in S. baicalensis According to Different Diseases

Baicalin and baicalein are the main flavonoid components in the roots of *S. baicalensis*. They have the same pharmacological effects, such as the inhibition of apoptosis, hepatoprotection, and anti-diabetes properties [51,52], but there are also differences in the pharmacological activities of baicalin and baicalein. For example, baicalin has effects on gastric mucosa protection and anti-heart failure [53,54], and baicalein has neuroprotective and antiviral effects [54,55]. In the study, the impact of different drying methods on the active ingredients varies during fresh processing. The oven drying (60 °C) is more favorable for preserving baicalein and wogonin, while the shade drying and sun drying methods are favorable for preserving baicalin and wogonoside. Therefore, different drying methods can be chosen for the quality control of *S. baicalensis* depending on the differences of the diseases that are to be treated.

In addition, the result of the PCA shows that the sample (moisture content of 24.38%) that was processed by the sun drying method under fresh processing conditions is the best processing condition, and that the content of baicalin continued to increase as the moisture content decreased. We recommend setting a moisture content gradient from 24.38% to 12% in the future to find more favorable processing conditions if a quality control is required for diseases that are to treated with baicalin alone.

## 4. Materials and Methods

### 4.1. Chemicals and Plant Materials

HPLC-grade acetonitrile was purchased from Merck Co., Inc. (Darmstadt, Germany). HPLC-grade phosphoric acid and analytical-grade methanol were purchased from Tianjin Komiou Chemical Reagent Co., Ltd. (Tianjin, China). Four reference compounds (baicalein, baicalin, wogonoside, wogonin, HPLC ≥ 98%) were purchased from Shanghai Yuan ye Bio-Technology Co., Ltd. (Shanghai, China). Ultrapure water was generated using a You put Ultrapure Water System (Sichuan, China).

The fresh roots of two-year-old *S. baicalensis* plants were harvested in April 2020 from Tongchuan (Shaanxi, China). Additionally, they were identified by Professor Zongsuo Liang (Zhejiang Sci-tech University). The samples were divided into 16 batches after being cleaned of soil and impurities.

### 4.2. Sample Process

Fresh processing: The roots of the freshly harvested *S. baicalensis* plants were divided into 13 groups (Codes S1–S13). As shown in Table 3, different drying conditions were applied. The *S. baicalensis* roots were dried for different durations to achieve different moisture contents (Table 3). Then, the samples (S1–S13) were cut into 1–2-mm-thin slices, and the slices were dried in an oven at 60 °C until they were completely dry (moisture content ≤ 12.0%). All of the treatments were replicated three times (Figure 9).

Traditional processing: The newly collected roots of *S. baicalensis* plants were divided into three groups (Codes S14–S16). As indicated in Table 3, different drying conditions were used. The *S. baicalensis* roots were dried for different durations to achieve different moisture content values. The samples (S14–S16) were then softened by steam and sliced into 1–2 mm thin slices before being dried in an oven at 60 °C until they were totally dry (12.0%). All of the treatments were replicated three times.Shade drying: This process was performed in a dark room. The S2–S5 and S14 samples were placed in a well-ventilated, dark room with no direct sunlight. The room’s average temperature was 13.5 °C, and the relative humidity was 55 ± 5%.Sun drying: This procedure was carried out in the open. The S6–S9 and S15 samples were exposed to direct sunshine on the hardboard. The average temperature was 16.8 °C (from mid-late March to mid-April).Oven drying: The S10-S13 and S16 samples were dried at 60 °C using an electric constant-temperature blast-drying oven (DHG-9240A, Shanghai Jing Hong Laboratory Instrument Co., Ltd., Shanghai, China).

### 4.3. Preparation of Samples and Reference Standards

The dried samples were sieved through a Standard Chinese Pharmacopeia sieve-3 (355 ± 13 μm). A total of 0.2 g of each sample was accurately weighed, and then, they were placed in an erlenmeyer flask. The sample was subjected to ultrasonic extraction with 25 mL of 70% methanol for 40 min at room temperature. The supernatant was filtered through a 0.22 μm membrane. The samples were stored in a refrigerator at 4 °C before the HPLC-PDA analysis.

Appropriate amounts of each of the four reference standards of baicalein, baicalin, wogonoside, and wogonin were dissolved in methanol (99.8%) to obtain the desired final contents (2871.80, 1000.00, 657.50, and 100.00 µg·mL^−1^ of the mixed reference standard solution, respectively), then, the mixed solution was bottled and stored in a refrigerator at 4 °C.

### 4.4. Instrumentation and HPLC Conditions

A fingerprinting analysis of S. baicalensis was conducted using a Waters 1525 HPLC system (Waters Corp., Milford, MA, USA) that included a binary pump, a 2996 photodiode array detector, a column temperature controller, and a manual injector, and a binary pump. The data acquisition and system control were supported by Empower 2 software (Waters Corp., Milford, MA, USA). A Waters Symmetry C18 column (4.6 mm × 250 mm, 5 µm) was used for the chromatographic separation. The gradient elution of A (acetonitrile) and B (0.2% phosphoric acid solution) were utilized as follows: 0–20 min, 30–50% A; 20–30 min, 50–50% A; 30–50 min, 50–70% A; 50–60 min, 70–30% A. The column temperature was set to 30 °C, and the flow rate was set at 0.8 mL min^−1^. The chromatographic peak area (PA) was monitored via the HPLC-PDA detector at wavelengths of 277 nm.

### 4.5. Determination of Moisture Content in Sample

The moisture content was assessed using the drying method that was described in the fourth section of the Chinese Pharmacopeia 2020 edition [56]. The samples were that were used were from approximately 2 to 5 g, and they were spread in a flat weighing bottle and dried to constant weight. The thickness of the sample should be 5 mm. The bottle with samples was weighed and dried for 5 h at 100–105 °C. Then, the bottle was sealed with a cap and transferred to a desiccator. We kept it cool for 30 min, we weighed it accurately, and it was dried at the above temperature for 1 h. We allowed it to cool, and we weighed it until the weight difference between the two consecutive weighing steps did not exceed 5 mg. The sample’s moisture content was calculated based on the weight loss.

### 4.6. Main Peak Identification and Sample Content Determination

The reference standard solution was injected into the HPLC system using the HPLC conditions indicated in Section 4.4, and the tR and PA of the reference standard were recorded. The active ingredients of all of the samples were identified by comparing the tR values to those of the reference standards. The contents of the active ingredients were calculated according to external standard methods.

### 4.7. Similarity Evaluation and Statistical Analysis

The HPLC fingerprint similarity analysis (SA) was evaluated using Similarity Evaluation System for Chromatographic Fingerprint (Version 2012) software which was developed and recommended by the Chinese Pharmacopoeia Committee. The HCA and PCA of all of the samples were performed using the Statistical Program for Social Sciences (SPSS) 26 software (SPSS Inc., Cary, NC, USA).

SA: The HPLC data of 16 batches of *S. baicalensis* were imported into the “Similarity Evaluation System for Chromatographic Fingerprinting in Chinese Medicine (2012 version)”. The S1 sample was set as the reference spectrum, and the median method was used. The time window width was 0.1, a multi-point calibration was performed, and the control spectrum was generated.HCA: HCA is a multivariate analysis method that provides visual information about raw data [57]. The relative PAs of the 18 common peaks of 16 batches of the *S. baicalensis* samples were imported into IBM SPSS 26 software to perform a cluster analysis with the between-group connection method and using the Euclidean square distance as the classification basis.PCA: PCA is a commonly used multivariate analysis method, and it is usually employed in HPLC fingerprinting research [58,59,60]. In this study, the PA of 14 unique peaks (A3, A4, A7 (baicalin), A8, A9, A10, A11, A12, A13 (wogonoside), A14, A15, A16 (baicalein), A17, and A18 (wogonin)) in 16 groups of samples, as described above, was submitted to SPSS 26 for the PCA. The best principal components were defined as those with eigenvalues λ > 1, and 4 principal components were extracted. The eigenvalues of the first four principal components were 4.563, 3.242, 2.304, and 1.476, and the cumulative contribution rate was 82.747% (Appendix A), which satisfied the requirement of the cumulative percentage of variance (CPV) > 70–85% for the PCA. Therefore, the four principal components contained most of the information in the variables. The variance contribution of the principal components was used as the weight, and the first four main components were used to obtain a total score (F1–4, F) for assessing the quality of 16 batches of samples. Then, the comprehensive evaluation was calculated. The higher the F value was, the better the quality of the samples were that we measured.

## 5. Conclusions

In this study, the effect of the fresh processing method on the roots of *S. baicalensis* was investigated for the first time, along with the evaluation of three different drying methods of *S. baicalensis*. The four main active components in *S. baicalensis* were quantified through the HPLC, and overall, the 18 components were evaluated by using the SA of the HPLC fingerprinting. Finally, the best processing method for the roots of *S. baicalensis* was determined by HCA and PCA. Our results showed that the fresh processing method yielded a relatively excellent preservation of wogonoside, baicalein, and wogonin rather than that which was yielded with the traditional processing method, and under certain drying and moisture content conditions, the fresh processing method can also better preserve the baicalin. In addition, we found that the sun drying way was more favorable for maintaining baicalin and wogonoside, and the oven drying way (60 °C) was more profitable for preserving baicalein and wogonin (*p* ≥ 0.05). Finally, combining the results with the PCA, we concluded that the sun drying method (moisture content: 24.38%) during fresh processing was the most beneficial post-harvest processing method. Collectively, the study’s results will provide a reference for quality control of *S. baicalensis* in the original cultivated area.

## Figures and Tables

**Figure 1 molecules-27-08302-f001:**
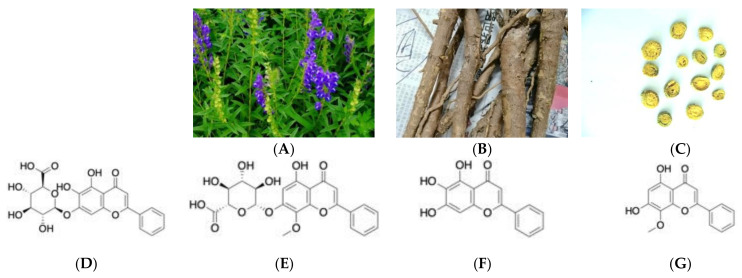
Pictures of the total plant (**A**), TCM crude drugs (**B**), and TCM prepared slices (**C**) of *Scutellaria baicalensis*; the structure of Baicalin (**D**), Wogonoside (**E**), Baicalein (**F**), and Wogonin (**G**).

**Figure 2 molecules-27-08302-f002:**
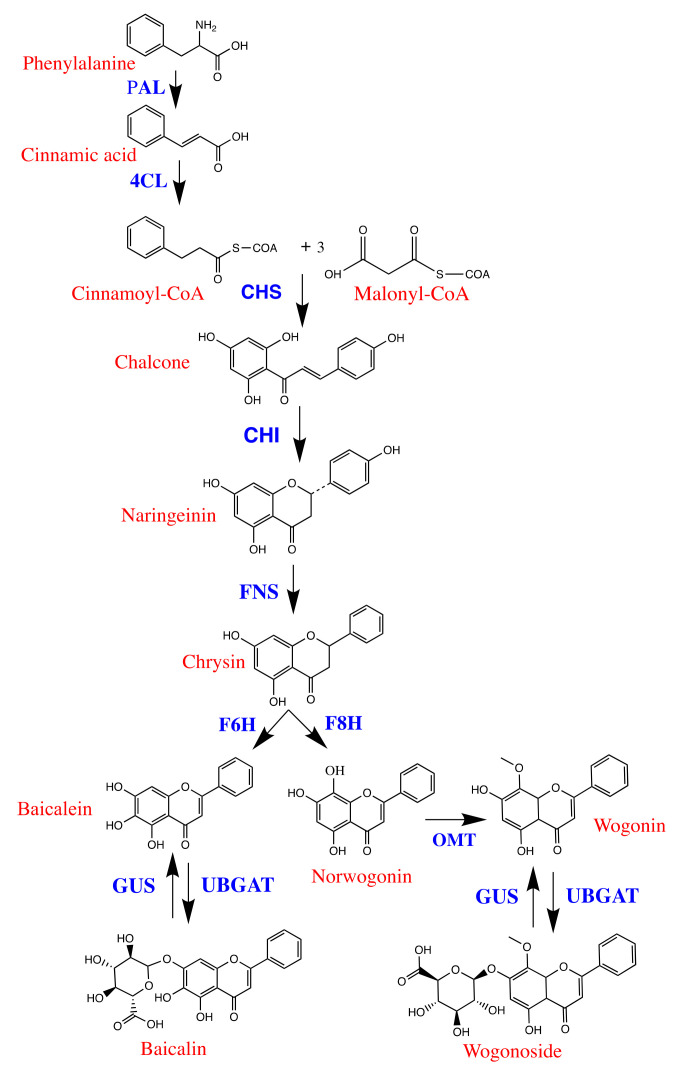
The flavonoid biosynthesis in roots of *S. baicalensis*. PAL: phenylalanine ammonia-lyase; 4CL: 4-coumarate CoA ligase; CHS: chalcone synthase; CHI: chalcone isomerase; FNS: flavone synthase; F6H: flavone 6-hydroxylase; F8H: flavone 8-hydroxylase; OMT: O-methylatransferase; UBGAT: UDP-glucuronosyltranferase; baicalein 7-O-glucuronyltransferase GUS: Glucuronidase.

**Figure 3 molecules-27-08302-f003:**
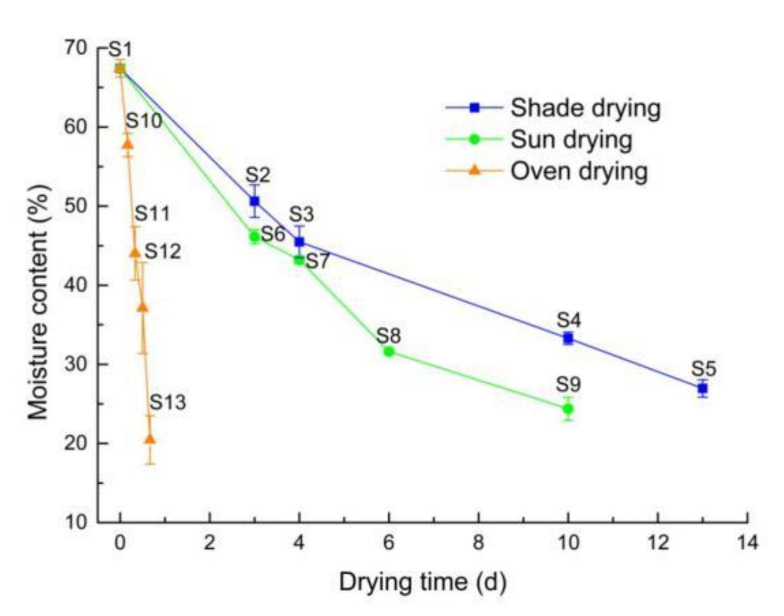
Drying curve of *S. baicalensis* under different drying condition.

**Figure 4 molecules-27-08302-f004:**
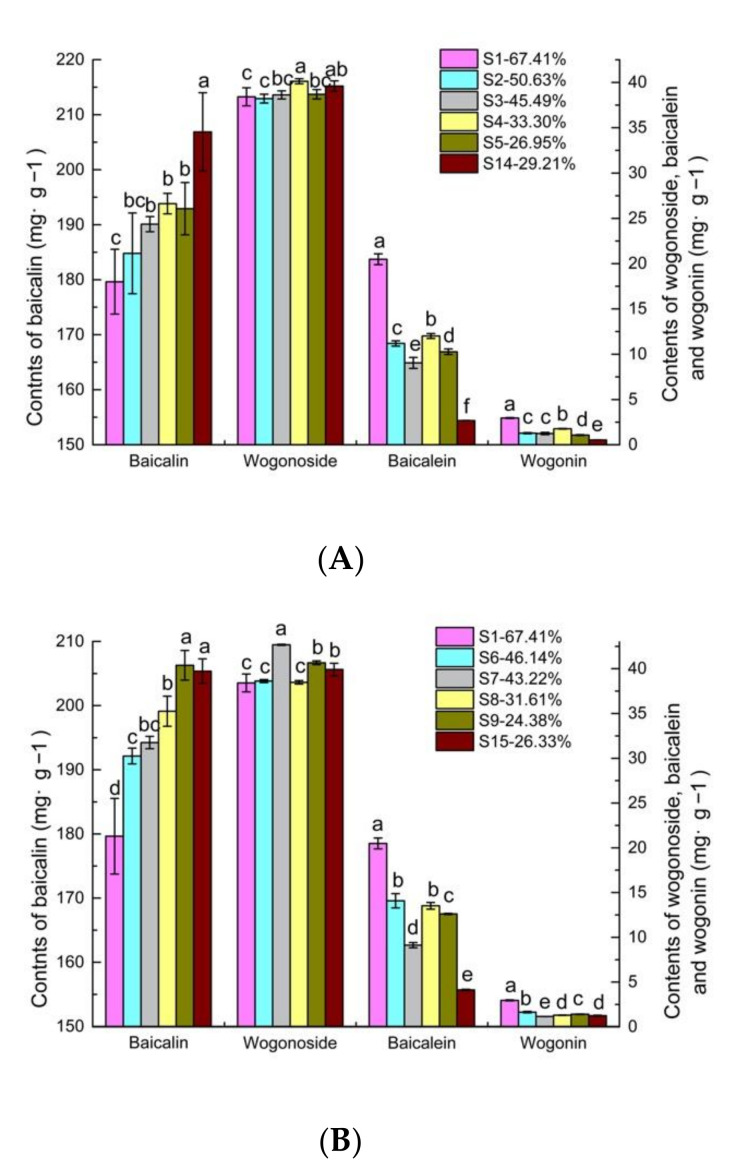
Changes in ingredient contents under different processing methods. (**A**) Shade drying; (**B**) sun drying; (**C**) oven drying. Note: *p* < 0.05. a–f: differences between groups.

**Figure 5 molecules-27-08302-f005:**
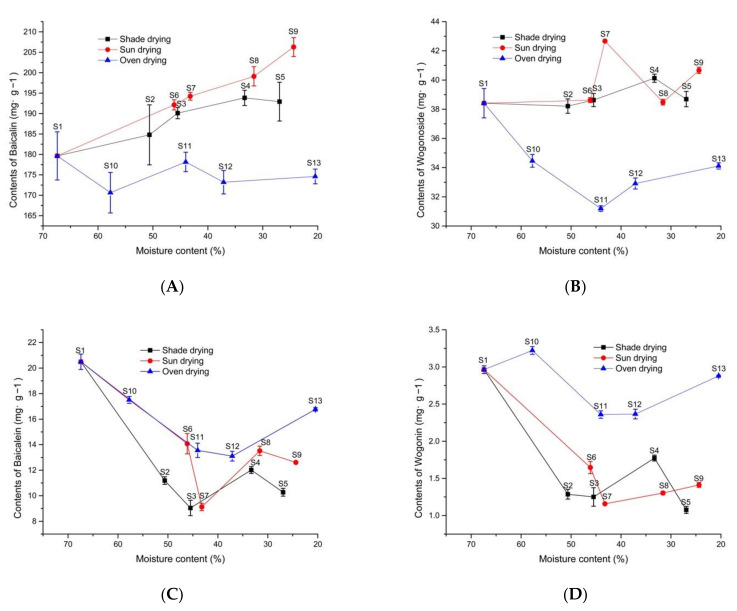
Changes in the content of ingredients in *S. baicalensis* under different drying conditions. (**A**) baicalin, (**B**) wogonoside, (**C**) baicalein, and (**D**) wogonin.

**Figure 6 molecules-27-08302-f006:**
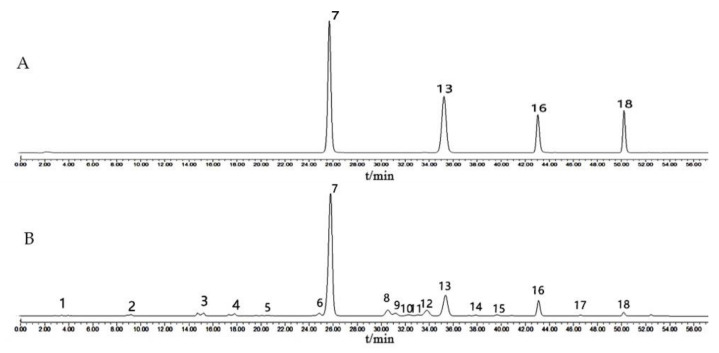
Typical chromatograms of standard mixed solution (**A**) and samples (**B**). Peak 7—Baicalin; peak 13—Wogonoside; peak 16—Baicalein; peak 18—Wogonin.

**Figure 7 molecules-27-08302-f007:**
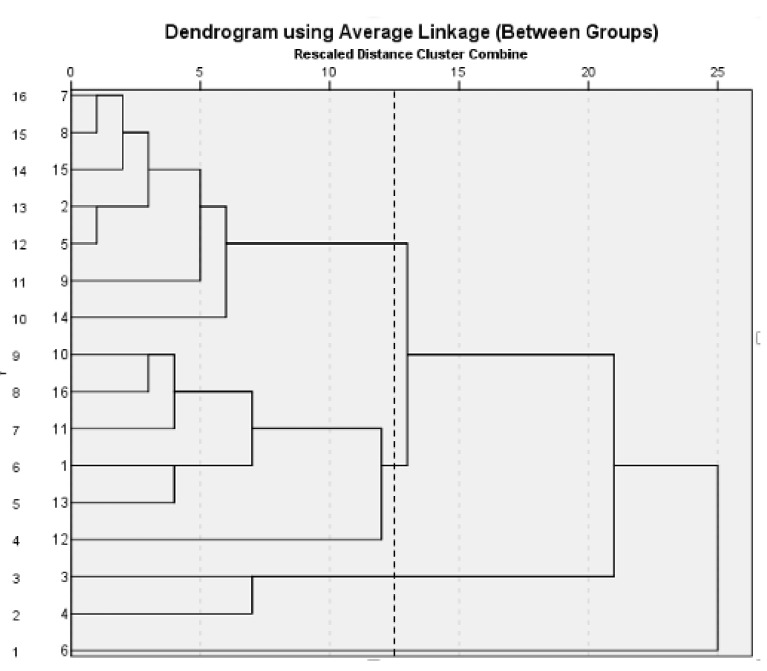
The cluster dendrogram of all active ingredients of 16 batches of different processing methods of *S. baicalensis*.

**Figure 8 molecules-27-08302-f008:**
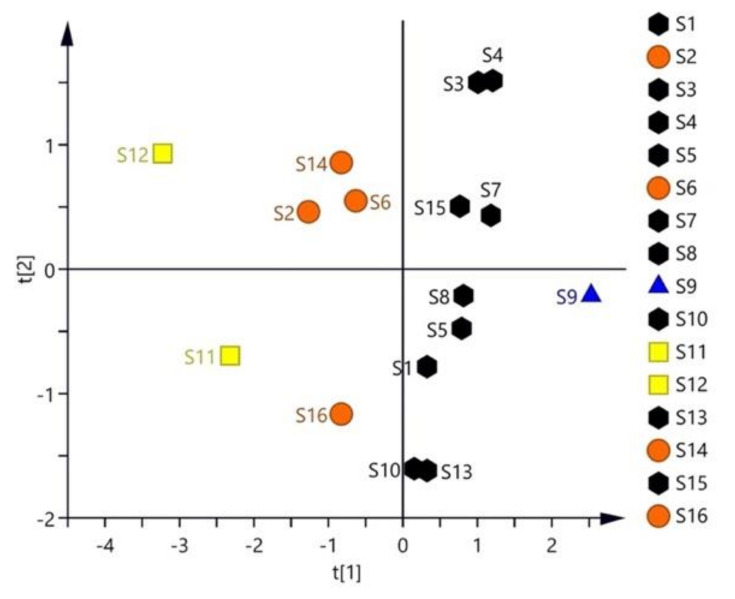
The 2D score plot of all active ingredients of 16 batches of different processing methods of *S. baicalensis* with the principal components.

**Figure 9 molecules-27-08302-f009:**
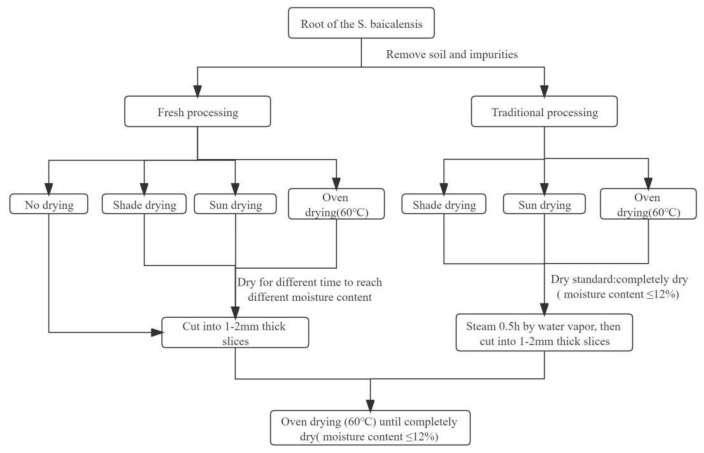
The process flow of *S. baicalensis* roots.

**Table 1 molecules-27-08302-t001:** Contents of four active ingredients in S1–S16.

Code	Baicalin (mg/g)	Wogonoside (mg/g)	Baicalein (mg/g)	Wogonin (mg/g)
S1	179.639 ± 5.902 *^,#,&^	38.413 ± 1.002 *^,#,&^	20.488 ± 0.602 *^,#,&^	2.964 ± 0.052 *^,#,&^
S2	184.800 ± 7.346 *	38.214 ± 0.491 *	11.188 ± 0.298 *	1.286 ± 0.065 *
S3	190.103 ± 1.380 *	38.627 ± 0.446	9.039 ± 0.601 *	1.250 ± 0.124 *
S4	193.837 ± 1.859 *	40.129 ± 0.280	12.013 ± 0.271 *	1.773 ± 0.039 *
S5	192.921 ± 4.745 *	38.695 ± 0.521	10.275 ± 0.306 *	1.075 ± 0.047 *
S6	192.139 ± 1.229 ^#^	38.624 ± 0.170 ^#^	14.078 ± 0.795 ^#^	1.646 ± 0.081 ^#^
S7	194.235 ± 0.0962 ^#^	42.670 ± 0.063 ^#^	9.125 ± 0.291 ^#^	1.157 ± 0.010 ^#^
S8	199.114 ± 2.351 ^#^	38.480 ± 0.192 ^#^	13.520 ± 0.370 ^#^	1.302 ± 0.023
S9	206.283 ± 2.314	40.666 ± 0.212	12.612 ± 0.071 ^#^	1.412 ± 0.033 ^#^
S10	170.637 ± 4.966	34.463 ± 0.438	17.507 ± 0.269 ^&^	3.221 ± 0.051 ^&^
S11	178.155 ± 2.369 ^&^	31.197 ± 0.191 ^&^	13.555 ± 0.563 ^&^	2.359 ± 0.050 ^&^
S12	173.209 ± 2.867 ^&^	32.908 ± 0.376 ^&^	13.103 ± 0.382 ^&^	2.366 ± 0.065 ^&^
S13	174.598 ± 1.800 ^&^	34.113 ± 0.210	16.768 ± 0.129	2.879 ± 0.010 ^&^
S14	206.869 ± 7.122	39.602 ± 0.577	2.674 ± 0.019	0.541 ± 0.010
S15	205.369 ± 1.920	39.917 ± 0.695	4.129 ± 0.063	1.249 ± 0.054
S16	166.339 ± 2.001	33.830 ± 0.201	16.247 ± 0.221	4.327 ± 0.038

* *p* < 0.05 vs. S14, ^#^
*p* < 0.05 vs. S15, ^&^
*p* < 0.05 vs. S16.

**Table 2 molecules-27-08302-t002:** The values of the principal component, comprehensive scores, and ranking of *S. baicalensis* conducted by different processing.

Num	F1	F2	F3	F4	F
S9	2.144	211.635	12.580	1.461	51.938
S7	1.152	125.916	15.531	−0.163	32.076
S8	0.878	94.554	−13.613	0.673	20.016
S15	1.139	59.529	16.717	−0.203	16.888
S5	0.375	65.166	−3.551	0.957	14.731
S3	3.017	17.630	37.239	−0.611	11.130
S4	3.549	−23.350	49.405	−0.176	3.860
S13	−1.921	8.517	−0.606	1.734	1.430
S1	−1.648	−17.347	34.456	0.676	1.186
S10	−2.472	−7.315	9.168	1.455	−0.838
S16	−2.832	−64.017	0.444	0.443	−15.629
S6	−0.613	−105.592	38.514	−0.955	−18.418
S2	−0.180	−70.514	−22.723	−1.021	−20.236
S14	2.300	−56.046	−49.216	−0.421	−20.374
S11	−2.914	−110.530	−48.219	−0.737	−34.560
S12	−1.974	−128.240	−50.537	−3.112	−38.988

**Table 3 molecules-27-08302-t003:** Different processing treatment methods.

Sample Code	Drying Conditions		Time/d	Moisture Content before Cutting/%	Moisture Content after Drying/%
S1	Fresh processing	No drying	0	67.41	4.12
S2		Shade drying	3	50.63	4.44
S3			4	45.49	5.30
S4			10	33.30	4.73
S5			13	26.95	4.52
S6		Sun drying	3	46.14	4.00
S7			4	43.22	8.77
S8			6	31.61	6.49
S9			10	24.38	5.79
S10		Oven drying (60 °C)	1/3	57.74	6.07
S11			5/6	44.03	4.98
S12			4/3	37.13	5.70
S13			11/6	20.46	6.04
S14	Traditonal processing	Shade drying	35	29.21	6.54
S15		Sun drying	25	26.33	6.29
S16		Oven drying (60 °C)	2	32.38	5.82

## Data Availability

Not applicable.

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
