# Peer review of "Post-Harvest Processing Methods Have Critical Roles in the Contents of Active Ingredients of Scutellaria baicalensis Georgi"

_molecules, 2022, doi:10.3390/molecules27238302_

Round 1

Reviewer 1 Report

Comments

This paper describes quantitative analyses of typical flavonoids of Scutellaria baicalensis and claims merits of new flesh processing method for the plant’s roots.

The proposal for new processing methods of traditional medicine is meaningful. And the HPLC method for the quantitative analyses of Scutellaria flavonoids is seemed to be appropriate.

On the other hand, this reviewer regret that I can’t agree with acceptance of this manuscript. The scientific novelty and verification methods are so limited. In my opinion, the results are lack of number of attempts for each method and verification of conditions for obtained amounts of flavonoids.

This reviewer considers that it is necessary to clear noted below (1-8).

1The advantages of the new processing method are considered to be small, and the superiority of the new method over the conventional method has not been objectively evaluated. However, I afraid of my narrow-mindedness, therefore it would be necessary to have this evaluated by TCM experts.

2. It seems that this manuscript discusses about only one set of test samples. It is needed to process more than one material, ex. samples from various makers and various harvesting area, in the same way and compare them.

3. There seems to be a great deal of variability regarding the conditions under which it dries (concerning not dry, shade, and sun). The usefulness of this processing method can be scientifically demonstrated only by showing what factors determine the amounts of flavonoids by controlling temperature and light intensity and measuring them under different established correct conditions.

4P.13 4.7. Similarity Evaluation and Statistical Analysis: Please describe in detail the method of statistic examination.

5. Figure 1: The chemical structures (configurations) of glucuronic acid moieties of baicalin (GluA-1) and wogonoside (L-configuration) are incorrect. Furthermore, I recommend remaking it to clear.

6. I think that figures 2 and 6 are not needed.

7Figure 4 and Table 1 showed the identical data? Only the unit of vertical axis of baicalin was different. Therefore, I think these need to be made a little clearer.

8. There are typos and description errors in the manuscript. It is recommended to check again carefully. Italic style for Scutellaria baicalensis (lines 30, 63). Capitals letters in sentences (lines 38, 364), for example.

Author Response

Dear reviewers:

        We have carefully considered the suggestion of Reviewer and made some changes. We have tried our best to improve and made some changes in the manuscript.

Responds to the reviewers' comments:

Review#1

Main comment:

Point 1:The advantages of the new processing method are considered to be small, and the superiority of the new method over the conventional method has not been objectively evaluated. However, I afraid of my narrow-mindedness, therefore it would be necessary to have this evaluated by TCM experts.

Response 1:We appreciate it very much for this good suggestion, we will further validate the new method with pharmacological and metabolomic experiments if we have the opportunity to research the new processing method of the next Chinese herb. In addition, we also mentioned in the discussion section 3.3 Line 314-317 that we suggest further narrowing down the moisture scope to find a better solution for the fresh processing method of S. baicalensis.In our current experiments, the advantages of fresh processing are not so obvious to the preservation of the content of Baicalin and Wogonoside, but it is a great advantage for the accumulation of Baicalein and Wogonin. In addition, fresh cutting in the original cultivated area saves processing costs, time, and energy while maintaining the quality of the S. baicalensis Compared to traditional processing methods, it is still a good method for the processing of the S. baicalensi and can replace the traditional processing methods.

Point 2:It seems that this manuscript discusses about only one set of test samples. It is needed to process more than one material, ex. samples from various makers and various harvesting area, in the same way and compare them.

Response 2:Thank you for your comment. We have done pre-experiments on Scutellaria baicalensis from different origins in Tongchuan, Shaanxi, China, and Shangluo, Shaanxi, China, and on two different brands of Scutellaria baicalensis. We found that the origin and brand did not affect the processing aspect.

Point 3There seems to be a great deal of variability regarding the conditions under which it dries (concerning not dry, shade, and sun). The usefulness of this processing method can be scientifically demonstrated only by showing what factors determine the amounts of flavonoids by controlling temperature and light intensity and measuring them under different established correct conditions.

Response 3This is a very good and rigorous suggestion, thank you very much, and we will take your suggestion in our future research work. However, the purpose of our experiment was to simulate the actual production of Huangqin (the root of S. baicalensis) in the production area. We have tried our best to monitor the details of the experiment, such as the temperature and humidity of the experiment, and all the original records are available. In addition, during the actual mass production in the production area, herbs are collected from different regions and manufacturers, and it is difficult for each region and manufacturer to achieve the same environmental conditions, such as controlling the same natural light and outdoor temperature.

Point 4P.13 4.7. Similarity Evaluation and Statistical Analysis: Please describe in detail the method of statistical examination.

Response 4Thank you for your suggestion. The SA, HCA, and PCA method have been described in section 2.3 line 185-194, section 2.4 line 195-198, and section 2.5 line 209-221. And we have transferred the detail of SA、HCA、PCA methods to section 4.7 in the revised manuscript.

Line 394-398 SA: The HPLC data of 16 batches of S.baicalensis were imported into the "Similarity Evaluation System for Chromatographic Fingerprinting in Chinese Medicine (2012 version)". S1 was set as the reference spectrum, and the median method was used. The time window width was 0.1, multi-point calibration was performed, and the control spectrum was generated.

Line399-403 HCA: The relative PAs of the 18 common peaks of 16 batches of S. baicalensis samples were imported into IBM SPSS 26 software to perform cluster analysis with the between-group connection method and the Euclidean square distance as the classification basis.

Line404-418 PCA: In this study, the PA of 14 unique peaks(A3, A4, A7(Baicalin), A8, A9, A10, A11, A12, A13(Wogonoside), A14, A15, A16(Baicalein), A17, A18(Wogonin))in 16 groups of samples, as described above, was submitted to SPSS 26 for PCA. The best principal components were defined as those with eigenvalues λ>1, and 4 principal components were extracted. The eigenvalues of the first four principal components were 4.563, 3.242, 2.304, 1.476, and the cumulative contribution rate was 82.747% (Table S4), which satisfied the requirement of cumulative percentage of variance (CPV)>70-85% for PCA. Therefore the four principal components contained most of the information in the variables. The variance contribution of principal components was used as the weight, and the first four main components were used to obtain a total score (F1-4, F) for assessing the quality of 16 batches of samples. Then the comprehensive evaluation was calculated. The higher the F value, the better the quality of the samples measured.

Point 5Figure 1: The chemical structures (configurations) of glucuronic acid moieties of baicalin (GluA-1) and wogonoside (L-configuration) are incorrect. Furthermore, I recommend remaking it to be clear.

Response 5We are very sorry for our mistake and thank the reviewer for reminding us. We have corrected this mistake.

Point 6I think that figures 2 and 6 are not needed.

Response 6Thank you for the reviewer's suggestion. We have deleted figure 6 but Retained figure 2. Figure 2 was the biosynthetic route of the components in S. baicalensis, some of the content in part of the introduction and discussion has to be explained with the help of figure 2.

Point 7Figure 4 and Table 1 showed the identical data? Only the unit of vertical axis of baicalin was different. Therefore, I think these need to be made a little clearer.

Response 7Yes, the data in Figure 4 and Table 1 was identical. To show the differences in the composition of S. baicalensis more clearly, we made Figure 4. The left vertical axis is the unit of measurement of baicalin, and the right vertical axis is the unit of measurement of baicalein, wogonoside, and wogonin. In Figures, 4A, B, and C, the maximum values of wogonoside were 40.129, 42.670, and 38.413 mg/g respectively, while the maximum values of baicalin were 206.869, 206.283, and 179.639 mg/g. So the unit of lift vertical axis was different compared to the unit of the right vertical axis. In addition, the values of wogonoside and wogonin are more than ten times higher. To take into account the description of the content of the three components(baicalein, wogonoside, and wogonin)in the right vertical axis, we have tried our best to present the current solutions.

Point 8There are typos and description errors in the manuscript. It is recommended to check again carefully. Italic style for Scutellaria baicalensis (lines 30, 63). Capitals letters in sentences (lines 38, 364), for example.

Response 8We are very sorry for our mistake and thank the reviewer for reminding us. We have checked our manuscript carefully and corrected some problems. For example, Capital letters and Italic style, figure and table, and so on.

        Thank you for your precious comments and advice. Those comments are all valuable and very helpful for revising and improving our paper. We have revised the manuscript accordingly, and our point-by-point responses are presented above. We also sincerely hope that this revised manuscript has addressed all your comments and suggestions.

Reviewer 2 Report

Line 45: please add capital letter after the point (baicalin)

figure 1: please add letters (A, B, ...G) to the figures and chemical structures.

Line 47: "accounting for around 12%" w/w? please specify.

Line 49: "baicalein has a wider absorption window". The author should better explain this sentence. Towards what?

Line 68: please add "(TMC)" after writing Traditional Chinese medicine

Lines 98-100: explain acronyms (HPLC, SA, HCA, PCA)

Lines 104-107: the slower decrease in the moisture content with sun and shade drying methods compared to the oven method is expected. The authors should add a comment about this.

Section 2.2: the authors should indicate in the text which figure they are talking about (4A, 4B or 4C). For example, in line 112 they should write “From Figure 4A and Table 1, it can be seen that the content of baicalin was highest in S4.” However, from table 1 and Figure 4A, the content of baicalin is the highest in S14. Please explain better which data are shown in the figure (Fresh processing vs Traditional processing samples).

Lines 114-115: “The content of Wogonoside in S3, S4, and S5 was not significantly different from S14.” The authors should write that the content of Wogonoside in samples S1-S5 was not significantly different from S14.

Lines 115-116: check English “The contents of baicalein and wogonin were significantly found in the S1 were 6.66 and 4.48 times higher than in S14, respectively.”

Line 116: “It can be seen that the content of baicalin was THE highest in S9”

Lines 112-132: data show that S1 can be considered as the most advantageous method for wogonoside, baicalein and wogonin. The only exceptions are shade drying and sun drying methods for baicalin, that show higher concentrations. The authors should point this out in this section.

Figure 4: please indicate what letters a-f refer to.

Table 1: add the unit of measurement (mg/g?)

Lines 144-145: “According to the trend in Figure 5A, for the content of baicalin, sun drying (S1-S6-S7-144 S8-S9) > sun drying (S1-S2-S3-S4-S5) > oven drying (60°C) (S1-S10-S11-S12-S13).” Change “sun drying (S1-S2-S3-S4-S5)” with “shade drying (S1-S2-S3-S4-S5)”. In my opinion, S1 should not be included in the list of samples, since it can be considered as a starting point for each method. I would suggest writing “sun drying (S6-S7-144 S8-S9) > shade drying (S2-S3-S4-S5), etc”.

Line 147: how do the authors explain the decrease in baicalin content for the oven drying method? Possible degradation? Why is the content increasing with decreasing moisture content in sun and shade drying methods?

Lines 148-150: “For the content of wogonoside in the sample (Figure 5B), the sun drying method(S1-S6-S7-S8-S9) and shade drying (S1-S2-S3-S4-S5) method were greater than oven drying method(60°C) (S1-S10-S11-S12-S13).” Check English: does “were greater” mean “were better”? see also line 155.

Lines 148-154: How do the authors explain the behavior of wogonoside content (constant) with respect to that of baicalin (increase) in sun and shade drying methods?

Line 160: “was lower than in S1” change to “were lower”.

Figure 5: the authors should add labels to each point, to indicate which samples they are referring to. Is the y-axis correct? By looking at data reported in  Table 1, one would expect 10x higher values.

Line 176: “And the time”, delete “And”

Line 241: replace “we believe” with other words (i.e. it has been demonstrated, it is recommended, …)

Line 243: the authors should indicate which other studies.

Table 3: for a better readability the authors should add horizontal lines to divide the different drying methods (eg. A line should be added between S5 and S6, and so on).

Figure 10: please indicate the S1 method.

Line 366: substitute ug with µg.

Lines 372-374: remove “a binary pump, a 2996 photodiode array detector, a column temperature controller, and a manual injector”

Lines 377-378: 0-20min, 30%-50% A; 20-30min, 50%-50% A; 30-50min, 50%-70% A; 50-60min, 70%-30% A.

Lines 382-384: could you describe the method?

Line 387: “HPLC conditions indicated in section 2.4”, do the authors mean Section 4.4?

The authors write that “sun drying way was more favorable for maintaining baicalin and wogonoside, and the oven drying way (60) was more profitable for preserving baicalein and wogonin”.  However, data show that this is true only for baicalin and in a minor way for wogonoside, with very similar results to S1. Concerning baicalein and wogonin, S1 showed the best results. It seems that no drying (S1) is the best method, also in terms of energy/time/money/speed. The authors should add a comment on this.

Author Response

Dear reviewers:

We have carefully considered the suggestion of Reviewer and make some changes. We have tried our best to improve and made some changes in the manuscript.

Responds to the reviewers' comments:

Review#2

Point 1 . Line 45: please add capital letter after the point (baicalin)

Response 1: We are very sorry for our mistake and thank the reviewer for reminding us. We have corrected the mistake.

Point 2 . figure 1: please add letters (A, B, ...G) to the figures and chemical structures.

Response 2: Thank you very much for your advice, I have added letters to the figures and chemical structures in figure 1.

Point 3 . Line 47: "accounting for around 12%" w/w? please specify.

Response 3: Thank you very much for your advice, we have corrected the problem. Line 49 "accounting for around 0.1g/g(Ratio in raw material)"and the relevant references have been added(reference 20.21).

Point 4 . Line 49: "baicalein has a wider absorption window". The author should better explain this sentence. Towards what?

Response 4: This suggestion is very good and I have added the relevant content in lines 52-56 “Baicalin has a wide window of absorption, and the degree of absorption of baicalin in all parts of the stomach and intestine is significantly better than that of baicalin. Baicalein is only moderately absorbed in the stomach and largely unabsorbed in the small intestine and colon; baicalein is well absorbed in the stomach and small intestine and to some extent in the colon.”

Point 5 . Line 68: please add "(TMC)" after writing Traditional Chinese medicine

Response 5: Thank you very much for your advice. I have added "(TMC)".

Point 6 .Lines 98-100: explain acronyms (HPLC, SA, HCA, PCA)

Response 6: We have explained acronyms. Line 110 HPLC: High performance liquid chromatography. Line111 SA: Similarity analysis. HCA:Hierarchical Cluster Analysis. PCA: Principal Component Analysis.

Point 7 . Lines 104-107: the slower decrease in the moisture content with sun and shade drying methods compared to the oven method is expected. The authors should add a comment about this.

Response 7: Thank you very much for your advice. we have added comments in lines 117-120 of revision manuscript.“The oven drying(60℃) method has a relatively high compared to the shade drying and sun drying method. In addition, the oven drying(60℃) method has a relatively stable heating environment, such as temperature, humidity, and light, so the moisture content of the sample dried by the shade drying and sun drying method decreases more slowly.”

Point 8: Section 2.2: the authors should indicate in the text which figure they are talking about (4A, 4B or 4C). For example, in line 112 they should write “From Figure 4A and Table 1, it can be seen that the content of baicalin was highest in S4.” However, from table 1 and Figure 4A, the content of baicalin is the highest in S14. Please explain better which data are shown in the figure (Fresh processing vs Traditional processing samples).

Response 8: Thank you very much for your good advice.We have made modifications in lines 112、116、122. In addition, we also have revised line 112 to “In combination with Figures 4A and Table 1, for the samples(S2, S3, S4 and S5) processed by fresh processing,the content of baicalin is the highest in S4.however, its content was still reduced by 6.30% compared to the content of S14 processed by Traditional processing .

Point 9: Lines 114-115: “The content of Wogonoside in S3, S4, and S5 was not significantly different from S14.” The authors should write that the content of Wogonoside in samples S1-S5 was not significantly different from S14.

Response 9: Thank you for the reviewer's suggestion, however, as can be seen in Figure 4A, S1 and S2 are both labeled with the letter c, and S14 is labeled with the letter ab, so there is a significant difference between S1, S2 and S14.

Point 10:Lines 115-116: check English “The contents of baicalein and wogonin were significantly found in the S1 were 6.66 and 4.48 times higher than in S14, respectively.”

Response 10: Thank you for the reviewer's suggestion, we have revised this section to “The contents of baicalein and wogonin in the S1 were 6.66 and 4.48 times higher than in S14, respectively.”

Point 11: Line 116: “It can be seen that the content of baicalin was THE highest in S9”

Response 11: Thank you for the reviewer's suggestion, we have corrected the mistake.

Point 12:Lines 112-132: data show that S1 can be considered as the most advantageous method for wogonoside, baicalein and wogonin. The only exceptions are shade drying and sun drying methods for baicalin, that show higher concentrations. The authors should point this out in this section.

Response 12: This is a very good suggestion.We have carefully consider your suggestion in combination with the experimental data, we could conclude that S1 is more favourable for the preservation of baicalein. However, for wogonoside and wogonin, we can clearly find in Figure 4B that S7 and S9 are more favourable for wogonoside than S1, and that S10 is favourable for wogonin than S1 in Figure 4C. Therefore, we have added “S1 can be considered as the most advantageous method for baicalein” in Line165-166.

Point 13:Figure 4: please indicate what letters a-f refer to.

Response 13: Thank you for the reviewer's suggestion,We have added the“a-f refer to differences between groups.” in line 153 of the revision mamuscript.

Point 14:Table 1: add the unit of measurement (mg/g?)

Response 14: We are very sorry for our mistake and thank the reviewer for reminding us. we have added it (mg/g) in table 1.

Point 15:Lines 144-145: “According to the trend in Figure 5A, for the content of baicalin, sun drying (S1-S6-S7-144 S8-S9) > sun drying (S1-S2-S3-S4-S5) > oven drying (60°C) (S1-S10-S11-S12-S13).” Change “sun drying (S1-S2-S3-S4-S5)” with “shade drying (S1-S2-S3-S4-S5)”. In my opinion, S1 should not be included in the list of samples, since it can be considered as a starting point for each method. I would suggest writing “sun drying (S6-S7-144 S8-S9) > shade drying (S2-S3-S4-S5), etc”.

Response 15: Thank you very much for your advice. I have made the appropriate revisions. For example,line158-159“According to the trend in Figure 5A, for the content of baicalin, shade drying (S6-S7-S8-S9) > sun drying (S2-S3-S4-S5) > oven drying (60°C) (S10-S11-S12-S13). ”

Line162-164“For the content of wogonoside in the sample(Figure 5B), the sun drying method(S6-S7-S8-S9) and shade drying (S2-S3-S4-S5) method were better than oven drying method(60°C) (S10-S11-S12-S13).”

Line169-171“For the content of baicalein and wogonin in samples(Figure 5C、5D), the oven drying(60℃)method (S10-S11-S12-S13) were better than the shade drying method (S2-S3-S4-S5) and the sun drying method (S6-S7-S8-S9).”

Point 16:Line 147: how do the authors explain the decrease in baicalin content for the oven drying method? Possible degradation? Why is the content increasing with decreasing moisture content in sun and shade drying methods?

Response 16: The question have been explained in the discussion section in manuscript. Line 274-287“The roots of S. baicalensis are physiologically active for some time after harvesting, and prolonged high-temperature stress (40°C) inhibits the activity of phenylalanine ammonia-lyase (PAL) and consequently affects the accumulation of baicalin [46]. The content of baicalin and wogonoside is significantly reduced in samples dried in an oven (60°C) drying. During the oven drying (60°C) process, the internal heat could not be dissipated in time with the extension of drying time, resulting in continuous heating inside the roots. That creates robust conditions for a series of complex enzymatic oxidation, polymerization, and other biochemical reactions. Some studies have shown that the enzymatic (endogenous enzyme glucuronidase, GUS) reaction of baicalin and other glycosides is accelerated in a hot and humid environment, resulting in the enzymatic degradation of baicalin and wogonoside to baicalein and wogonin, which contain phenolic hydroxyl groups in adjacent positions and are easily oxidized to quinone derivatives and appear green [47, 48]. In addition, many phenolic and alcoholic hydroxyl groups in the flavonoid structure are unstable and easily destroyed at high temperatures. the study has come to the same conclusion [49].”

Point 17:Lines 148-150: “For the content of wogonoside in the sample (Figure 5B), the sun drying method(S1-S6-S7-S8-S9) and shade drying (S1-S2-S3-S4-S5) method were greater than oven drying method(60°C) (S1-S10-S11-S12-S13).” Check English: does “were greater” mean “were better”? see also line 155.

Response 17: Thank you very much. The “were greater” mean “were better”,We have checked the Line 148-150 and Line155 and revised its.

Line 162-164“For the content of wogonoside in the sample(Figure 5B), the sun drying method(S6-S7-S8-S9) and shade drying (S2-S3-S4-S5) method were better than oven drying method(60°C) (S10-S11-S12-S13)”.

Line169-171“For the content of baicalein and wogonin in samples(Figure 5C、5D), the oven drying(60℃)method (S10-S11-S12-S13) were better than the shade drying method (S2-S3-S4-S5) and the sun drying method (S6-S7-S8-S9).”

Point 18: Lines 148-154: How do the authors explain the behavior of wogonoside content (constant) with respect to that of baicalin (increase) in sun and shade drying methods?

Response 18:The enzymes of the medicinal plant roots remain physiologically active under shade drying and natural drying conditions. As the biosynthetic pathways of baicalin and wogonoside were different. As seen in figure 2, chrysin was converted to baicalin by the F6H, to norwogonin by F8H, and norwogonin was converted wogonin by OMT. After the baicalein and wogonin were converted to baicalin and wogonoside by UBGAT. Compared to baicalin, the biosynthesis of wogonoside was one step longer, and it was regulated by different enzymes, so the variation in wogonoside content differs from that of baicalin.

Point 19:Line 160: “was lower than in S1” change to “were lower”.

Response 19: Thank you very much for your advice.we have revised it.

Point 20: Figure 5: the authors should add labels to each point, to indicate which samples they are referring to. Is the y-axis correct? By looking at data reported in  Table 1, one would expect 10x higher values.

Response 20:We are very sorry for our mistake and thank the reviewer for reminding us. We have re-made figure 5 and modified the Y-axis, then also added the labels to each sample.

Point 21: Line 176: “And the time”, delete “And”

Response 21: Thank you very much for your advice. we have deleted “and”

Point 22: Line 241: replace “we believe” with other words (i.e. it has been demonstrated, it is recommended, …)

Response 22: Thank you very much, we have revised the word"we believe" to "it has been demonstrated"

Point 23: Line 243: the authors should indicate which other studies.

Response 23: We are very sorry for our mistake and thank the reviewer for reminding , we have added the reference 37and 38.

37.Li,L.;Zhang,C.;Yu,D.;Ma,Y.;Tian,G.;Wang,Y.;Huang,W., Study on Habitat Processing Method of Scutellaria baicalensis. Chinese Journal of Experimental Traditional Medical Formulae 2011,17,(8),1-3

38.Zhang, X.;Zhang, L.; He,K.;Liu,J.;Xu,H.;Liang,Z.; Study on fresh processing of Scutellaria baicalensis based on HPLC fingerprintand multi-index quantification. Chinese Traditional and Herbal Drugs,2021,52,(15).4552-4560.

Point 24:Table 3: for a better readability the authors should add horizontal lines to divide the different drying methods (eg. A line should be added between S5 and S6, and so on).

Response 24:We have added horizontal lines to divide the different drying methods in Table 3.

Point 25:Figure 10: please indicate the S1 method.

Response 25: Thank you very much for your advice. We have supplemented the method of S1 in figure10.

Point 26:Line 366: substitute ug with µg.

Response 26: We are very sorry for our mistake and thank the reviewer for reminding us, we have revised the mistake.

Point 27:Lines 372-374: remove “a binary pump, a 2996 photodiode array detector, a column temperature controller, and a manual injector”

Response 27: we have deleted “a binary pump, a 2996 photodiode array detector, a column temperature controller, and a manual injector”

Point 28: Lines 377-378: 0-20min, 30%-50% A; 20-30min, 50%-50% A; 30-50min, 50%-70% A; 50-60min, 70%-30% A.

Response 28: We have revised it to “0-20min, 30%-50% A; 20-30min, 50%-50% A; 30-50min, 50%-70% A; 50-60min, 70%-30% A.” in lines 369-370 of the revised manuscript.

Point 29:Lines 382-384: could you describe the method?

Response 29: Thank you very much for your advice, I have supplemented the relevant methods in section 4.5 line375-382“The samples were taken approximately 2 to 5 g and spread in a flat weighing bottle dried to constant weight. The thickness of the sample should be 5 mm. The bottle with samples was weighed and dried for 5 hours at 100-105°C. Then the bottle is sealed with a cap and transferred to a desiccator. Keep it cool for 30 minutes, weigh it accurately, and it was dried at the above temperature for 1 hour. Allow it to cool, and weigh it until the weight difference between the two consecutive weighings does not exceed 5mg. The sample's moisture content was calculated based on weight loss.”

Point 30:Line 387: “HPLC conditions indicated in section 2.4”, do the authors mean Section 4.4?

Response 30: Thank you very much for your advice,I have corrected it.

Point 31:The authors write that “sun drying way was more favorable for maintaining baicalin and wogonoside, and the oven drying way (60℃) was more profitable for preserving baicalein and wogonin”.  However, data show that this is true only for baicalin and in a minor way for wogonoside, with very similar results to S1. Concerning baicalein and wogonin, S1 showed the best results. It seems that no drying (S1) is the best method, also in terms of energy/time/money/speed. The authors should add a comment on this.

Response 31: We appreciate it very much for this good suggestion, and we have added “S1 can be considered as the most advantageous method for baicalein” in Line165-166.However, for wogonoside and wogonin, we can find in Figure 4B that S7 and S9 are more favourable for wogonoside than S1 and that S10 is favourable for wogonin than S1 in Figure 4C. In the principal component analysis, baicalin and wogonoside have a high percentage of peak area and are assigned high values, and the F-score of S1 is also very low when taken together. The content of baicalin was used as the only quantitative indicator component in Part I of the Chinese Pharmacopoeia(2020 version). Therefore, S1 (no drying) isn’t the best method.

     I would like to thank the reviewers again for your precious comments and advice.We have revised the manuscript accordingly,and our point-by-point responses are presented above.We also sincerely hope that this revised manuscript has addressed all your comments and suggestions. 

Round 2

Reviewer 1 Report

I agree to accept this manuscript.

Author Response

Dear reviewer:

      Thank you for the reviewers’ comments concerning our manuscript entitled “Post-harvest processing methods have critical roles in the contents of active ingredients of Scutellaria baicalensis Georgi” (ID: molecules-2007821). 

Reviewer 2 Report

lines 49-50: at the beginning it was 12%, now it is 0.1g/g. The authors should clarify this difference.

line 121: "The oven drying(60°C) method has a relatively high compared to" check english. It has a high what?

line 130: "is the highest in S4.however," check punctuation, leave a black space after the dot and use capital letters. Check the entire manuscript.

Author Response

Dear reviewers:

         We have carefully considered the suggestion of Reviewer and make some changes.  We have tried our best to improve and made some changes in the manuscript.

Responds to the reviewers' comments:

Review#2

Point 1:lines 49-50: at the beginning it was 12%, now it is 0.1g/g. The authors should clarify this difference.

Response 1: We appreciate it very much for this good suggestion. According to the references we have cited, there were differences in baicalin in S. baicalensis extracted by different extraction methods. And there were differences to baicalin in S. baicalensis produced from various locations ,  according to our known data the range of baicalin is around 8%-13%, so we took an approximate value of 10%. In addition, according to last suggestion that "accounting for around 12%" w/w, I have converted “10%” to “0.1g/g”.

Reference:

1.Li-Weber, M., New therapeutic aspects of flavones: the anticancer properties of Scutellaria and its main active constituents Wogonin, Baicalein and Baicalin. Cancer Treat Rev 2009, 35, (1), 57-68.

2.Wang, H. Z.; Yu, C. H.; Gao, J.; Zhao, G. R., Effects of processing and extracting methods on active components in Radix Scutellariae by HPLC analysis. China Journal of Chinese Materia Medica 2007, 32, (16), 1637-1640.

3.Yang, L.; Liu, D.; Feng, X.;Cui, S,;Yang, J.;Tang, X.; He, X.; Liu, J.; Hu, S. Determination of Flavone for Scutellaria baicalensis from Different Areas by HPLC. China Journal of Chinese Materia Medica 2002,03,9-12.

Point 2: line 121: "The oven drying(60°C) method has a relatively high compared to" check english. It has a high what?

Response 2: We are very sorry for our mistake and thank the reviewer for reminding us. We forgot to write the word “temperature”, we have added the word “temperature” in line 121:"The oven drying(60°C) method has a relatively high temperature compared to the shade drying and sun drying method."

Point 3 : line 130: "is the highest in S4.however," check punctuation, leave a black space after the dot and use capital letters. Check the entire manuscript.

Response 3: We are very sorry for our mistake and thank the reviewer for reminding us. We have corrected the mistake.In addition, we also have checked the entire manuscript carefully and corrected some problems: the Capital letters in lines 133、153、327、260、366 and Italic style in line 408, Tense lines in 18 and Sentence coherence in lines 153、172、340 and Singular Plural in lines 25、338.

         Thank you for your precious comments and advice. We have revised the manuscript accordingly, and our point-by-point responses are presented above. We also sincerely hope that this revised manuscript has addressed all your comments and suggestions.